# The Effects of Visual Complexity and Task Difficulty on the Comprehensive Cognitive Efficiency of Cluster Separation Tasks

**DOI:** 10.3390/bs13100827

**Published:** 2023-10-09

**Authors:** Qi Guo, Yan Chen

**Affiliations:** School of Art Design and Media, East China University of Science and Technology, Shanghai 200030, China; y81220078@mail.ecust.edu.cn

**Keywords:** visual complexity, task difficulty, cognition efficiency, visualization design

## Abstract

Cluster separation is required to perform multi-class visual statistics tasks and plays an essential role in information processing in visualization. This cognition behavioral study investigated the cluster separation task and the effects of visual complexity and task difficulty. A total of 32 college students (18 men and 14 women, with ages ranging from 18 to 25 years; mean = 21.2, SD = 3.9) participated in this study. The observers’ average response accuracy, reaction time, mental effort, and comprehensive cognitive efficiency were measured as functions of three levels of visual complexity and task difficulty. The levels of visual complexity and task difficulty were quantified via an optimized complexity evaluation method and discrimination judgment task, respectively. The results showed that visual complexity and task difficulty significantly influenced comprehensive cognitive efficiency. Moreover, a strong interaction was observed between the effects of visual complexity and task difficulty. However, there was no positive linear relationship between the mental effort and the complexity level. Furthermore, two-dimensional color × shape redundant coding showed higher cognitive efficiency at low task difficulty levels. In contrast, the one-dimensional color encoding approach showed higher cognitive efficiency at increased task difficulty levels. The findings of this study provide valuable insights into designing more efficient and user-friendly visualization in the future.

## 1. Introduction

The information interface requires complex information input and output processes as the leading information exchange channel between humans and information systems [1]. With the rapid development of digital technology and increasing information complexity, information interfaces tend to display information overload. Information overload occurs when the amount of input to an interface exceeds the processing capacity of the user. Selecting an appropriate information interface design based on different information dimensions and data types can alleviate cognitive overload caused by information overload and enable users to make accurate decisions. Currently, serious safety issues frequently arise from the imbalance between information interface design and user cognitive mechanisms. In recent years, more than 50% of civil aircraft accidents worldwide have been caused by pilots’ biased information judgment on the flight control interface, leading to poor mission decisions [2]. Therefore, exploring human visual cognitive behavior patterns is a prerequisite for effective information interface design, which has attracted widespread attention.

In general, effective information interface design mainly depends on the precision of visual representation [3] and the rationality of visual task design [4]. Visual representation mainly refers to the ability of information to be accurately recognized by users. Visualization designers have collaborated with perception researchers in the last few years to bring forth improvements. Influential studies include those of Eastman and Bertin [5], Cleveland and McGill [6], Lewandowsky and Spence [7], Rensink and Baldridge [8], Szafir [9], and Shen et al [10]. They have demonstrated that people could quickly and efficiently understand the content presented in large data sets if the information is translated into a visual graphical language that fits the visual cognitive model [11,12,13,14]. Scatterplots are a common form of visual representation. Scatterplots can simultaneously display multiple data sets to help to compare correlations and determine trends in each data set. Researchers have studied the ability of viewers to quickly and accurately assess trends [8,15]. However, these studies did not consider in depth how different forms of visual representation affect statistics judgments and different representation complexity levels. They mainly experimentally obtained the cognitive ordering of different forms of information graphs and the differences in the visual performance of specific graphs (e.g., scatter plots) under different correlation conditions. Additionally, a visual task refers to the user’s ability to obtain the desired information through a specific task design. Therefore, visual task design is essential for an effective interface, and visual presentations should be organized according to different tasks and purposes.

Visual complexity is the main influencing factor for the adequate perception of visual representations in an information interface. Different visual representations exhibit different levels of complexity within the same task. Understanding the definition of complexity and the factors affecting visual complexity is crucial for studying task performance [16]. Many studies have investigated the definition of complexity from different perspectives. For example, Heaps and Handel [17] attempted to define visual complexity from the interface perspective. In their experimental study, the independent variable was set to three different numbers of images, each with a different complexity. According to their experimental results, visual complexity was defined as the difficulty in describing the visual interface, a finding that is consistent with Palumbo’s study [18]. Deng et al. [19] proposed that visual complexity refers to the number of visual presentations and the level of information specifics, while others divided visual complexity into several visual features and design complexity [20]. Zhang [21] proposed that complexity describes the richness of visual presentations, with more visual components indicating higher complexity. Within the same task, different visual presentations might show different levels of complexity [22]. From the above findings, visual complexity is directly related to the information content presented in the visualization interface. Moreover, the measurement of visual complexity needs to be in line with humans’ subjective perception of complexity. The smallest constituent units for information visualization are visual variables, such as color, shape, and texture. According to the composition theory, the overall complexity of the visualization interface is composed of the complexity of these smallest feature units, meaning that the quantitative calculation results of visualization complexity can be obtained. Following a previous study [23], this study focuses on visual complexity as subjectively perceived by observers from the perspective of information target recognition. Our study defines visual complexity as the difficulty and familiarity of comprehending information, which is classified by setting different encoding forms.

Empirical studies have been conducted to investigate the visual features that influence visual complexity or can be used to measure complexity [24,25,26,27,28,29], aiming to develop complexity quantification methods. However, no consensus has yet been reached. Some studies divided visual complexity into different dimensions; interface complexity was measured by calculating the visual size complexity, local density complexity, grouping complexity, and alignment complexity of visual interface representations [30,31,32,33,34]. In contrast, other studies treated visual complexity as a primary configuration [23,35,36]. For example, the minor component units in the information interface are composed of visual variables in visual representations, such as color, shape, location, etc. According to the information composition theory [37], the complexity of these minor visual units constitutes the overall complexity of the interface. Therefore, visual complexity can be quantitatively calculated. Suo [38] constructed a visual integration complex tree to evaluate interface complexity. Using this method, the complexity of the information is assessed by building a visual integration complexity tree, where the number of nodes represents the upper limit of all visual content that a user may encounter while performing a visual task.

Visual complexity belongs to the design layer in the information interface, whereas task difficulty belongs to the task structure layer. According to the cognitive load theory [39,40], task difficulty is an important aspect contributing to cognitive load [41], and task difficulty is positively related to cognitive load. Higher-difficulty tasks are related to higher cognitive load and lower cognitive efficiency [42]. Task difficulty has been shown to affect search behavior and the performance outcomes of visual interfaces, such as the reaction time spent on the search results interface [43]. Therefore, the difficulty of specific tasks must be considered in visual design to control the overall cognitive load. Cluster statistics tasks are increasingly used in daily visualization instances, including mean identification [44], pattern discrimination [45], and magnitude summarization [46] for two or more clusters presented in real time in the information interface. Being able to distinguish different clusters clearly and compute statistical results quickly is required for subsequent information analysis. This entails the practical separation of clusters. The cluster separation task usually involves finding all the different classes and performing visual statistical calculations. Classes are groups of data points with the same label, which may be visually separated or overlapping, depending on how they are grouped. Assuming that different classes are displayed in different colors, many other design codes can influence this cognitive task, and users may have difficulty distinguishing two classes that are close to or overlap with each other if they are displayed in very similar colors. Sedlmair et al. [47] studied the class separation task by listing intra-class factors (e.g., density and anomaly) and inter-class factors (e.g., density and variance of the split) that affect finding, verifying, and matching clusters on scatters.

However, task difficulty and visual complexity are two factors that can be controlled during the information interface design stage. Notably, the factors used in most of the studies mentioned above were considered separately, either in terms of task difficulty or visual complexity. In contrast, studies have rarely investigated the combined or interaction effect of complexity and task difficulty on cluster categorization tasks and visual statistics task performance. In addition, most relevant experimental studies have used observers’ reaction times and accuracy levels as performance metrics without examining how task difficulty and interface complexity affect observers’ combined cognitive efficiency.

Feature selection, wholeness, and attention are involved in visual statistics tasks. Feature selection is broadly defined as focusing attention on various attributes of an object [48]. Most work on feature selection has been conducted using visual search tasks, in which observers can select individual groups of data defined by these features [49,50,51]. To reduce the computational load, our brain utilizes specific perceptual heuristics based on certain laws [52]. For example, the visual system is susceptible to symmetry [53], and there may be an obligatory, low-level interaction between symmetry and objecthood [54]. Our visual system also utilizes statistical laws to efficiently compress and process the information and then "weigh" the details of a single object, such as a book on a shelf, in favor of more detailed representations, a process known as ensemble encoding [55]. Visual features such as color, orientation, size, and spatial location are the basis for multiple integrated encoding [56]. Previous studies have also shown that it is unnecessary to continuously deploy attention to objects in a display to compute an overall description. As a result, we can predict that visual statistics tasks, whether averaging, correlation, etc., can be performed without requiring the viewer’s attention to any individual data point.

In summary, cluster separation is a crucial factor for performing multi-class visual statistics tasks and plays an essential role in information processing on visualization. People’s ability to efficiently extract statistical information from digital interfaces directly impacts their safety and user experience. Although previous studies have been conducted on visual complexity and visual representations, few studies have explored how task difficulty and interface complexity affect statistical task performance and what levels of task difficulty and interface complexity facilitate visual statistical decisions. Therefore, the present study explores the effects of task difficulty and visual complexity on information statistics performance. The study aims to evaluate comprehensive cognitive performance at task difficulty and interface complexity levels and analyze how visual representations affect visual cognitive behavior. Furthermore, the possible interaction effect between task difficulty and complexity is explored, and visual strategies appropriate for visual statistics are discussed.

## 2. Methods

### 2.1. Observers

A total of 32 college students (18 men and 14 women, aged 18 to 25 years old; Mean = 21.2, SD = 3.9) were included in this study. All observers were recruited through an online sign-up system. They were given the option of either USD 10 in cash or an equivalent gift as payment for taking part in the study, which required 30 min or less to complete. The observers had normal or corrected vision (20/25 visual acuity or better) without color blindness or weakness and reported no history of neurological, psychiatric, or emotional disorders. Before the experiment, the observers were required to fill in relevant information, including their name, age, and major and whether they had experimental experience in visual statistic tasks, and they were informed of the experiment rules and procedures. In the experiment, observers would be replaced if their results were more than two standard deviations beyond the average. Based on this criterion, no observers were excluded from the analysis in this experiment.

In order to familiarize each observer with the experimental task, a pilot test was conducted before the formal experiment. After the pilot experiment, observers confirmed that they understood the task thoroughly and were informed that the duration was between 30 and 50 min. In addition, a 5-min rest period was allowed after each sub-task to prevent observers from experiencing visual fatigue.

### 2.2. Apparatus

The experiment was carried out in the ergonomics laboratory at the East China University of Science and Technology. The Minolta Chroma Meter CS-200 was used to optimize color values and luminance, and the viewing distance was set and maintained at 55 cm from the screen. Stimuli were displayed on a 23.5 inch LCD monitor (Dell u2414 h) at a resolution of 24.6 cm high × 35.8 cm wide, using a refresh rate of 60 Hz. The display brightness was 92 cd/m^2^.

All experiments are developed in Javascript (ECMAScript 2020) and run on Google Chrome (http://www.google.cn/chrome/, accessed on 27 September 2023). And the jsPsych library was used to control the experimental tasks and the timeline of data collection.

### 2.3. Experimental Design

#### 2.3.1. The Visual Complexity Level Classification

##### Calculation of Visual Complexity

In this study, visual complexity was calculated in three steps. 

The first step involved establishing a complexity evaluation metric hierarchy. Each visualization was broken down into individual components based on the complexity evaluation metric hierarchy. From top to bottom, the hierarchy included the interface, visual structures, visual units, and encoding modalities.

In the second step, the complexity of each hierarchy level was scored based on the degree of understanding and degree of familiarity. The scoring values and descriptions are displayed in Table 1.

The third step was to calculate the total interface complexity. The complexity score of a hierarchical level was calculated using the sum of the complexity scores of its subordinate levels. The final complexity score for each level was determined based on the expert authority degree, the concentration degree, the dispersion degree, and the coordination degree.

(1) The expert authority degree $S_{R}$: $S_{R}$ is generally determined by two factors, namely expert familiarity ($S_{f}$) and the degree of understanding ($S_{u}$). The calculation formula was
(1)$$S_{R}=\sigma_{1}S_{f}+\sigma_{2}S_{u},\sigma_{1}+\sigma_{2}=1$$

(2) The concentration degree of expert scoring $\bar{C}$: The concentration degree of expert scoring was assessed by evaluating the weighted average of the complexity, with larger weighted averages indicating the higher complexity of the corresponding level. The calculation formula was
(2)$$\bar{C}=\sum_{j=1}^{5}C_{j}m_{ij}/q,(i=1,2,\ldots \ldots n)$$
where $C_{j}$ is the total complexity of a given metric (1, 2, 3, 4, 5). $m_{ij}$ represents the number of scores of the evaluation complexity i for the corresponding level j. q represents the number of experts. n is the number of the evaluation complexity. 

(3) The dispersion degree of the score $K_{i}$: The dispersion of the scores was reflected by the standard deviation, which indicated the dispersion of the overall complexity of an indicator, with smaller values suggesting lower dispersion. The calculation formula was
(3)$$K_{i}={\left(\sum_{j=1}^{5}m_{ij}{\left(C_{j}-\bar{C}\right)}^{2}/q-1\right)}^{1/2}$$

(4) Coefficients of variation $V_{i}$ and coordination coefficients ω for complexity scores:$$V_{i}=\bar{K_{i}}/\bar{C_{i}}$$
(4)$$\omega =\frac{12}{q^{2}(n^{3}-n)-q\sum_{k=1}^{q}\eta_{k}}\sum_{i=1}^{n}{\left(R_{i}-\bar{R}\right)}^{2}$$
where $V_{i}$ represents the coordination degree of the scorers’ estimation of the evaluation complexity i, with smaller values indicating higher coordination; ω represents the coordination degree of the overall evaluated complexity, with larger values indicating greater coordination; $R_{i}$ is the evaluation complexity i rank sum; $\bar{R}$ is the mean value of the rank sum of all evaluation complexities; and $\eta_{k}$ is the correction coefficient.

##### The Visual Complexity Level Classification Results

The visual complexity of the clusters was mainly reflected in how the visual units were encoded, especially the grouping of visual variables formed by each scatter. Different encoding forms, such as one-dimensional and multidimensional redundant encoding, multidimensional orthogonal encoding, and the presence of visual interference terms, lead to different complexities. Considering the various potential factors affecting the complexity of the interface, the scatter plot visualization interface was designed with different encoding methods, as shown in Table 2. The scatterplot visualization interfaces were drawn separately for each visual unit encoding method. The size of each interface was 1024 px × 768 px, with a scatter plot in the center of the interface and legends in the lower left and lower right corners. The scatterplots also included 50 scatter points per class, distributed in a 400 px × 400 px area. The color encoding used in the current study followed the study from Szafir [9], using red (CMYK = 45%, 97%, 99%, 15%) and blue (CMYK = 75%, 45%, 0%, 0%) as the two target scatterplot classes and a color with 2ND (significantly different) from the red brightness as the interference color (CMYK = 7%, 68%, 63%, 0%). The shape encoding included solid circles and hollow triangles with varying contour characteristics and topological properties. A fast Poisson disk sampling method [57] was used to avoid scatter point overlapping so as not to interfere with the observers’ overall perception of the scatter plot. The examples of eight different types of displays are shown in Figure 1.

A total of 52 scorers were recruited to score the complexity of each level in the eight interfaces. Since this study mainly focused on information visualization interfaces with analysis statistics, the scorers mainly consisted of interface designers, interface users, and experts in the field of information processing. Moreover, the complexity values of visual structures and visual units were set to constant values to minimize the influence of visual structures and visual units on the experimental results. Therefore, this experiment’s complexity scores were mainly determined via the encoding modalities. The content shown in Figure 2 was displayed on the computer. The scorer was then required to fill in a simple scale (0–5) to rate their understanding and familiarity with the display interface. Then, they were asked to score the complexity of each of the eight interface levels based on the content displayed.

Based on the complexity score calculation process described above, the eight displays were classified into three categories, namely low complexity, medium complexity, and high complexity, as shown in Table 3.

#### 2.3.2. Task Difficulty Level Classification

##### The Discrimination Method

More than three classes are usually integrated into practical scatterplot applications. However, in the actual cognitive process, the task was divided into multiple layers, with each layer containing only two separate tasks with two classes. Therefore, the task difficulty was set by distinguishing the average difference between two clusters of points. The task difficulty levels were obtained using the classical method commonly used in vision science. More specifically, the discrimination task was used to measure the perceptual precision and rank the difficulty of the separation task. In this approach, discrimination was measured based on just noticeable differences (JNDs) [58], i.e., the difference in stimulus intensity required to distinguish between the two groups of classes. In this case, the average height difference between the two groups of classes was used.

The same observers were assigned to this task. The demographics and apparatus did not vary from 2.1 and 2.2, and no observer was excluded from the analyses. Observers viewed two classes of scatterplots with different average heights, each class containing 100 black (CMYK = 82%, 78%, 76%, 59%) and red (CMYK = 27%, 97%, 100%, 0%) dots. Each dot was created using pseudo-random numbers taken from a Gaussian distribution, and the scatterplots were each of 6° vertical extent × 6° horizontal extent. Means were set to 0.5 of their extent, and standard deviations were set to 0.2 of this extent, in which the parameters of dots were designed from the simplified version of Rensink and Baldridge [8]. Assuming that the coordinates of any point in the scatterplot are (*x*, *y*), the y-coordinate *y*’ can be obtained using the following formula:(5)$$y’=rx+\sqrt{1-r^{2}}y\;r=0.2$$

Figure 3 shows the task stimuli. Any point greater than two standard deviations was excluded to avoid showing dots outside of the screen range, and a new dot was generated to replace it. Poisson sampling [59] was used in the generation algorithm to prevent overlaps so as not to interfere with observers’ overall perception of the scatterplots.

The average coordinates of the two classes were (*x*1, *y*1) for black dots and (*x*2, *y*2) for red dots, and the height difference was calculated via Δ*y* = *y*1 – *y*2, with the parameter Δ representing the pixel difference. The above parameters and attributes matched the displays derived from Michael Gleicher’s study [44].

The task difficulty level was measured by determining observers’ sensitivity to the difference between the Δ*y* of the two classes. The initial range of values for Δ*y* was [0 px, 200 px]. Δ*y* = 0 px indicated that the two classes had the same average height, and the observers could not distinguish between them. In contrast, Δ*y* = 200 px indicated that the two classes barely merged, and the plots were easily distinguished. The initial difference Δ*y* in the test was set at 50 px to reduce the experiment cost. In the experiment, observers were asked to report whether the black or red scatterplot had a higher average position. A step algorithm was used to adjust the difference Δ*y* after each observer responded on each trial. When observers made incorrect responses, the difference Δ*y* between the black and red scatterplots was increased by 5 px, making the next trial task easier. When observers correctly responded, the difference Δ*y* between the black and red scatterplots decreased by 3 px, making the task more difficult. Feedback was provided in each sub-condition. Observers made judgments until a steady-state accuracy of 75% was found for the JND, which was set as the moderate task difficulty level. In addition, when the steady-state accuracy was 55% and 95%, the corresponding Δ*y* were set as the high and low task difficulty levels, respectively.

Firstly, a 500 ms “+” sign was presented in the screen’s center to guide the observers. Then, the observers were shown two-class scatterplots in the center of the screen, with each containing black and red dots. The correlation of the two scatter plots was maintained at 0.2, and the initial differences in Δy value were set at 65 px. Subsequently, the observers were asked whether the black or red scatterplot showed a higher average position. If the average of the red scatter clusters was higher, the “A” key was pressed on the keyboard; if the average of the black red scatter clusters was higher, the “L” key was pressed. After each trial, the results were displayed on the screen, indicating to the observers whether they selected the correct color. Then, a blank screen was displayed for 500 ms before moving on to the next trial.

##### Task Difficulty Level Classification Results

Firstly, JND was calculated (the steady-state accuracy was 75%) when observers reported whether the black or red scatterplot showed a higher centroid. The customized experimental program Processing was used to record data, and variance analysis was performed in SPSS. The convergence algorithm obtained different response accuracy levels in this experiment [8]. The mean value of JND after convergence was taken as the final difference threshold.

Table 4 shows the mean performance results of the discrimination judgment trials. The analysis emphasized the effect size with a 95% confidence interval. An ANOVA analysis was first performed on the mean of the JND. The results showed no significant main effect on performance with or without experimental experience (F(3,24) = 8.21, *p* = 0.38 > 0.05; *η*2 =0.003). In addition, no significant difference in performance was found between different genders (μ_f_ = 60.1%, μ_m_ = 64.4%, *p* = 0.0838 > 0.05; *η*^2^ = 0.003). However, the main effect of the two-classes numbers reached statistical significance (F(2,161) = 5.65, *p* = 0.000 < 0.005; *η*^2^ = 0.007).

### 2.4. Experimental Hypotheses

This study used the following experimental hypotheses:(1)In the visual statistics task, the greater the task difficulty, the lower the overall cognitive efficiency;(2)The greater the task complexity, the lower the comprehensive cognitive efficiency;(3)Visual complexity and task difficulty significantly affect comprehensive cognitive efficiency;(4)Different visualization interfaces of the same complexity differ from each other in terms of cognitive efficiency due to their different encoding forms.

### 2.5. Experimental Procedure

The experiment consisted of 24 groups of within-subject trials (8 levels of encoding complexity × 3 levels of task difficulty). The observers were required to determine which of the two classes had a higher centroid in the scatters. The dependent variables for this experiment included the response accuracy (RA), response time (RT), mental effort (ME, obtained through the PAAS scale), and comprehensive cognitive efficiency (CE, obtained by importing RT, RA, and ME into the integrated cognitive efficiency formula developed by Paas and Nboer [60]). Observers were allowed to practice 10 training trials before their data were recorded.

The cognitive task of the experiment was consistent with the task difficulty discrimination method in The Discrimination Method section. Firstly, a 1000 ms “+” sign was presented in the center of the screen to guide the observers’ visual center. Then, the observers performed the cognitive task of determining which of the two classes had a higher average, with the encoded forms of the two scatter classes displayed in the lower left and lower right corners of the display, respectively. Observers were given as much time as needed to complete each task, although it was mentioned that accuracy was important. At the end of the trial, a blank screen was presented for 1000 ms to eliminate visual residuals. The observers then responded to the PAAS scale based on their mental perceptions from the previous trials, followed by a 2000 ms break before moving on to the next subset of trials. Next, 8 × 3 level sets were presented in random order, and the order of presentation of the three trials within each set was randomized. Each observer’s experiment lasted approximately 15 to 20 min.

### 2.6. Data Collection and Analysis

The RA, RT, ME, and CE were considered dependent variables when observers reported which classes had a higher average. The data were recorded by a custom experimental program and analyzed via ANOVA.

## 3. Experimental Results

For the RT analyses, only correct trials (7.3% error) were included. Moreover, cases with RTs greater than 2.5 median absolute deviations above or below the median were excluded from the analyses. The calculations were performed separately for each observer and condition (2.9%). Two-way ANOVA was used to analyze the effects of complexity and task difficulty (see Table 5).

### 3.1. Visual Complexity

The one-way ANOVA analysis results demonstrated significant differences in comprehensive cognitive efficiency between the different complexities, with a comprehensive cognitive efficiency main effect (F(2,2637) = 41.4060, *p* = 0.000 < 0.01, *η*^2^ = 0.001). In addition, the two factors of cognitive cost, namely RT and ME, showed significant differences, with an RT main effect (F(2,2637) = 24.141, *p* = 0.000 < 0.01, *η*^2^ = 0.001) and an ME (F(2,2637) = 64.402, *p* = 0.000 < 0.01, *η*^2^ = 0.001). However, the main effect of the cognitive benefit factor RA was not significant (F(2,2637) = 2.402, *p* = 0.0907 > 0.05, *η*^2^ = 0.001). The above results indicated that the cognitive cost is not worth the cognitive benefit for different complexities of the visual presentation.

Figure 4 shows the mean performances for the three levels of complexity. The comprehensive cognitive efficiency of the task gradually increased as the task complexity level decreased. Although visual complexity influences comprehensive cognitive efficiency, a significant difference in the three dimensions of cognitive efficiency measurement was not always observed. For example, there was little difference in RA between low and medium complexity, while the difference between them and the high complexity interface was more pronounced. However, the medium complexity visualization interface scored the highest in the mental effort dimension. The above shows no linear relationship between complexity and mental effort, as the highest ME was not observed at low complexity.

### 3.2. Task Difficulty

The results show the significant main effects of task difficulty in RA (F(3,2636) = 124.650, *p* < 0.001; *η*^2^ = 0.004), RT (F = (3,2636) = 87.545, *p* = 0.000 < 0.01, *η*^2^ = 0.005), ME (F = (3,2636) = 194.934, *p* = 0.000 < 0.01, *η*^2^ = 0.131), and the comprehensive cognitive efficiency CE (F = (3,2636) = 329.502, *p* = 0.000 < 0.01, *η*^2^ = 0.008). Figure 5 shows the mean task performance for the three levels of task difficulty. As the task’s difficulty decreased, the RA gradually increased from being almost indistinguishable to fully able to correctly discern. In contrast, the RT gradually decreased by about 2000 ms, and the level of mental effort gradually decreased with increasing task difficulty. Furthermore, the comprehensive cognitive efficiency gradually improved with increasing task difficulty. However, no significant change in ME was observed between the high and medium task difficulty level, which may be due to the visual saliency of the encoding.

### 3.3. Differences in Cognitive Efficiency between Different Encodings and Task Difficulty

A two-way ANOVA was conducted on eight different encodings and task difficulties. The results show significant differences between the encodings, with the main effects of RA (F(5,2634) = 4.660, *p* = 0.000 < 0.01, *η*^2^ = 0.001), RT (F(5,2634) = 23.185, *p* = 0.000 < 0.01, *η*^2^ = 0.001), and ME (F(5,2637) = 56.507, *p* = 0.000 < 0.01, *η*^2^ = 0.001). Notably, a strong interaction was observed between the different encoding and task difficulty, as indicated by the ME (F(15,2624) = 3.637, *p* = 0.000 < 0.01, *η*^2^ = 0.001) and CE (F(15,2624) = 3.060, *p* = 0.000 < 0.01, *η*^2^ = 0.003). 

Figure 6a shows the relationship between encodings and cognitive efficiency. The differences in cognitive efficiency within the same complexity but between different encodings reached statistical significance. In terms of comprehensive cognitive efficiency, the encodings, when ranked in descending order, were displays No. 1, No. 3, No. 5, No. 2, No. 7, No. 4, No. 8, and No. 6.

Subsequently, the comprehensive cognitive efficiency of No.1 (one-dimensional color encoding) and No. 3 (two-dimensional color × shape redundancy encoding) were further analyzed. Firstly, the difference in cognitive efficiency between the two was not significant (F(2,2637) = 1.469, *p* = 0.230 > 0.05, *η*^2^ = 0.007). However, the interaction between task difficulty and encoding was significant (F(2,2637) = 7.110, *p* = 0.000 < 0.01, *η*^2^ = 0.116). Figure 6b shows the CE of No.1 and No.3. under the low task difficulty condition, and the two-dimensional color × shape redundant encoding showed a higher cognitive efficiency. The one-dimensional color encoding showed higher cognitive efficiency at medium task difficulty.

The following conclusion shows that color is used as the primary coding. Color encoding dilutes the effects of other encoding. In Figure 6a, the colors are all within the odd-numbered ordinal numbers, meaning that there is a zigzag curve.

In addition, No. 2 (one-dimensional shape encoding) and No. 5 (two-dimensional color × shape orthogonal encoding) showed exceptional performance in the presentation complexity grouping. The cognitive efficiency of No. 2, which belongs to the low complexity level, was lower than that of No. 5, which belongs to the high complexity level, reflecting that visual complexity is only one aspect of evaluating cognitive efficiency. The visual complexity in a visual interface essentially reflects the complexity of how it is encoded. Color encoding is cognitively advantageous compared to other encodings [61]. Furthermore, the color may be psychologically preferred over the shape. Color encoding may interfere with non-color encoding [62,63]. In terms of RA, color encoding reduces the RA of size encoding by 29% and shape encoding by up to 43%. Therefore, display No. 2, featuring no color encoding distinction, showed a significant reduction in cognitive efficiency. On the other hand, although No.5 uses two encoding modes, the color is used as the primary encoding method. The impact of the other encoding is diluted by color encoding.

## 4. General Discussion

Visual complexity and task difficulty are two factors that can be controlled during the visualization design phase. Previous work mainly focused on complexity evaluation methods in GUIs [38,64,65,66] and the effect of visual complexity on the performance of specific search tasks [67,68]. However, few studies have investigated the combined or interactive effects of complexity and task difficulty on cluster separation task performance. Our work shows that visual complexity and task difficulty significantly affect the cognition efficiency of cluster separation tasks. The higher the task difficulty and complexity, the lower the visual statistics cognitive efficiency, which also depends on the type of visual encoding used. Moreover, task difficulty and complexity have an interactive effect.

Our research introduced a few caveats to existing design guidelines for visualization. For example, the perception and modeling of 3D scenes for games, autopilot display control interface and cabin operation interface, etc. The visualization in Figure 7a shows a map of a city with fixed (red-encoded) and moving (black-encoded) speed cameras, as well as speed limits (digitally encoded), with the maximum speed measured by each camera represented by a size encoding. The map has many points and is densely distributed, using multiple encoding methods, including complex shapes and numbers that are difficult to identify. If the user needs to know the distribution of fixed and mobile cameras in the city, as well as a comparison between the cameras’ number of alternating works in real time, an overly complex visualization can lead to a reduction in cognitive efficiency. Simplifying the location of each camera in the figure to a simple scatter in Figure 7b, and using only color encoding to distinguish between fixed and mobile cameras, can reduce the complexity of the interface and is more suitable for this task. 

The above design optimization aimed to maintain a certain level of RA. In some specific mission environments, such as combat, emergency repair, dispatch, etc., high demands are placed on RA, RT, ME, and CE. Color should be used as the primary encoding method in the interface design, and the cognitive difficulty of each task needs to be reduced to the minimum possible. Usually, task decomposition can be adapted to break down the complex task into multiple task units and improve the comprehensive cognitive efficiency of the task. Figure 7c shows the visualization of a power-monitoring interface. Each station to be monitored is displayed in the form of scatters on the map. Energy status is a priority indicator in monitoring, meaning that it is color encoded, with green used for normal, yellow for subhealth, and red for failure. Additionally, energy utilization (EBA) is a key indicator that needs to be monitored. If the number of indicators with EBA < 80% is greater than EBA < 40%, decision makers need to perform an immediate lockout and energy transfer. Due to the characteristics of the data, if EBA was directly encoded in the scatter diagram, the cognitive difficulty of the interface would increase. Therefore, indicator filtering is required to focus on monitoring objects and improve task cognition efficiency.

## 5. Conclusions and Future Work

This study examines the effects of visual complexity and task difficulty on the integrated cognitive efficiency of a clustering separation task. The visual complexity level and task difficulty were examined using an optimized complexity evaluation method and a discriminative judgment task, respectively. The present study revealed that visual complexity and task difficulty significantly influence comprehensive cognitive efficiency. A strong interaction effect was observed between the effects of visual complexity and task difficulty. Furthermore, a cognitive cost of the visual presentation was found for different levels of visual complexity. Redundant 2-D color × shape encoding showed higher cognitive efficiency at low levels of task difficulty, while the one-dimensional color encoding approach showed higher cognitive efficiency at increasing task difficulty levels.

This study mainly conducted experimental studies of two-class scatters, with many design attributes still needing to be investigated in multi-class scatters, including distribution type, point shape, point style, and dot cloud density. We also realized that there needed to be more experimental samples in our study. Lab studies with larger samples could investigate even more precise factor comparisons, such as individually encoded color, shape, or size. Additionally, despite visual complexity and task difficulty being the main controllable factors in visual statistics tasks, other factors, such as time pressure and other statistical tasks, affect visual cognitive efficiency. The current study provides a promising first step in exploring cognitive efficiency in visualizations, which could inspire other researchers to study more complex visual statistic processes and visualization designs.

## Figures and Tables

**Figure 1 behavsci-13-00827-f001:**
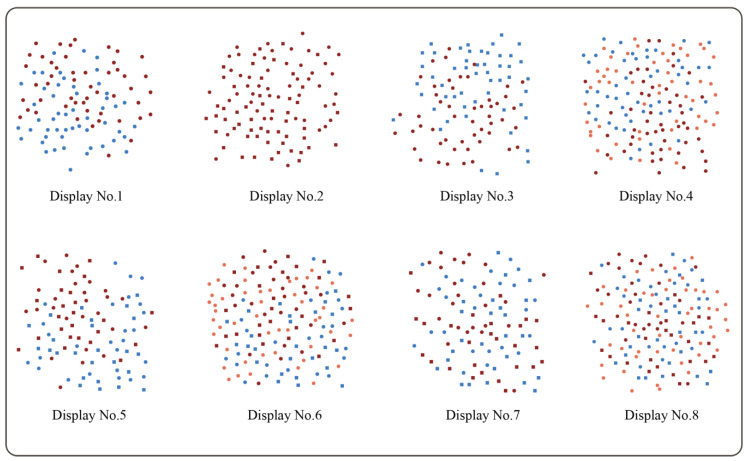
The samples used in this experiment.

**Figure 2 behavsci-13-00827-f002:**
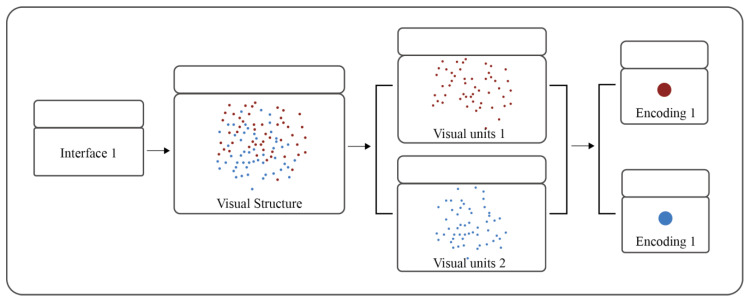
Schematic diagram of the scoring interface.

**Figure 3 behavsci-13-00827-f003:**
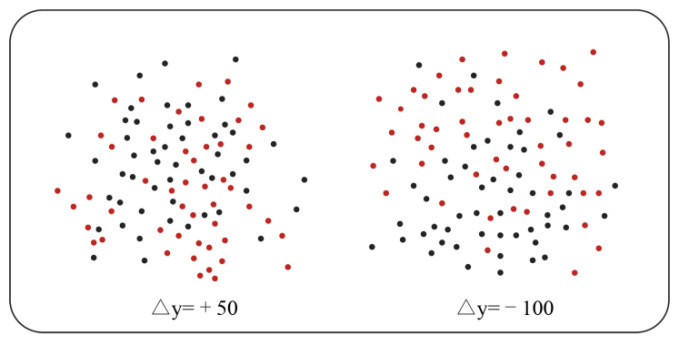
The experiment stimuli of task difficulty level classification.

**Figure 4 behavsci-13-00827-f004:**
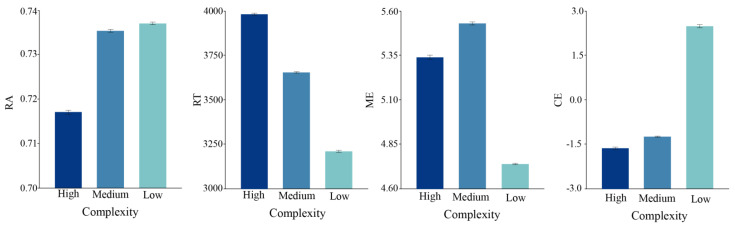
Main effects of complexity contrast on mean comparison task. Error bars indicate ±1 SEs.

**Figure 5 behavsci-13-00827-f005:**
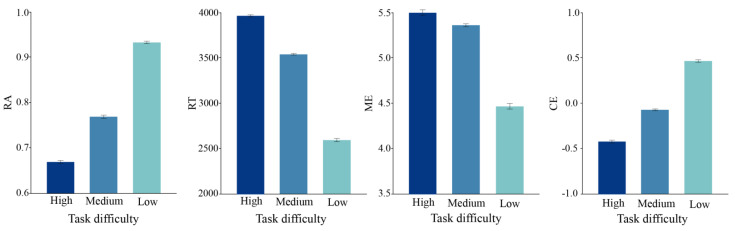
Main effects of task difficulty on average comparison task. Error bars indicate ±1 SEs.

**Figure 6 behavsci-13-00827-f006:**
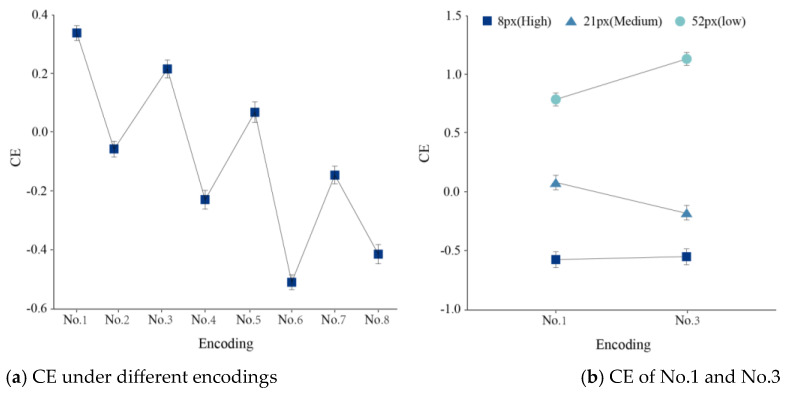
The relationship between encodings and cognitive efficiency.

**Figure 7 behavsci-13-00827-f007:**
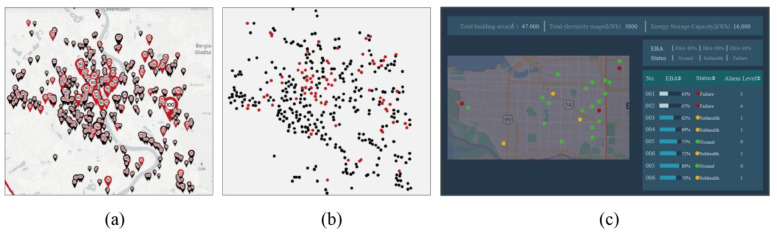
Visualization interface design examples.

**Table 1 behavsci-13-00827-t001:** The scoring of complexity of each hierarchy level.

Degree of Understanding (Refers to the Degree to Which the Evaluators Are Clear about the Logic of the Information Encoded in the Interface)		Degree of Familiarity (Refers to the Evaluators’ Familiarity with the Form of Information Encoded in the Interface)	
Very well understood	5	Very familiar	5
Well understood	4	Familiar	4
Relatively well understood	3	Relatively familiar	3
Generally well understood	2	Generally familiar	2
Not understood	1	Unfamiliar	1

**Table 2 behavsci-13-00827-t002:** The encoding modalities of different interfaces.

DisplayNo.	Encoding Forms	Target 1	Target 2	Interference
1	Color encoding	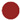	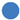	
2	Shape encoding	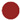	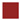	
3	Color * Shape (redundancy encoding)	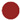		
4	Interference color encoding	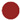	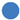	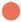
5	Color * Shape (orthogonal encoding)	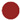 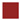	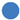 	
6	Color * Shape (orthogonal and interference encoding)	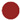 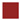	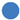 	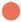
7	Shape * Color (orthogonal encoding)	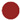 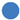	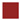 	
8	Shape * Color (interference encoding)	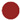 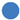	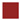 	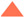

* is “and”.

**Table 3 behavsci-13-00827-t003:** Complexity score calculation results.

Display No.	$\bar{C}$	*K*	*V*	*ω*	Mean	Complexity Level
1	5.02	0.71	0.22	0.81	2.71	Low level
2	4.87	0.76	0.31	0.80	2.79	
3	4.98	0.89	0.25	0.76	2.81	
4	4.97	1.01	0.22	0.79	6.89	Medium level
5	5.01	0.83	0.31	0.76	12.31	High level
6	4.93	0.92	0.29	0.80	13.37	
7	5.07	0.72	0.22	0.77	13.29	
8	4.90	0.86	0.25	0.79	13.14	

**Table 4 behavsci-13-00827-t004:** Task difficulty level classification results (the mean of the JND).

Steady-State Accuracy	55%	75%	95%
The mean Δ*y*	8 px ± 1	21 px ± 2	52 px ± 2
Task difficulty level	High	Middle	Low

**Table 5 behavsci-13-00827-t005:** Two-way ANOVA of RA, RT, and ME in terms of complexity and task difficulty.

Sources	Variable	Sum of Squares (SS)	df	F	Sig.
Corrected model	RA	72.854 a	23	18.590	0.000
	RT	2,810,590,074.445 b	23	18.022	0.000
	ME	2202.952 c	23	43.141	0.000
	CE	1274.590 d	23	62.271	0.000
Task difficulty	RA	64.428	3	126.042	0.000
	RT	1,861,806,572.850	3	91.528	0.000
	ME	1454.553	3	218.383	0.000
	CE	982.542	3	368.024	0.000
Complexity	RA	3.970	5	4.660	0.000
	RT	786,042,613.761	5	23.185	0.000
	ME	627.285	5	56.507	0.000
	CE	251.201	5	56.454	0.000
Task difficulty * Complexity	RA	4.456	15	1.743	0.037
	RT	162,740,887.834	15	1.600	0.066
	ME	121.115	15	3.637	0.000
	CE	40.847	15	3.060	0.000
Error	RA	445.736	2616		0.000
	RT	17,737,760,706.191	2616		
	ME	5808.011	2616		
	CE	2328.048	2616		

a. Adjusted *R*^2^ = 0.133; b. adjusted *R*^2^ = 0.129; c. adjusted *R*^2^ = 0.269; d. adjusted *R*^2^ = 0.348. * is “and”.

## Data Availability

The datasets generated and/or analyzed during the current study are available from the first author on reasonable request.

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
