# Peer review of "The Effects of Visual Complexity and Task Difficulty on the Comprehensive Cognitive Efficiency of Cluster Separation Tasks"

_behavsci, 2023, doi:10.3390/bs13100827_

Round 1
Reviewer 1 Report (Previous Reviewer 1)
The authors have adequately addressed the points and suggestions provided in the last round of reviews. This has helped clarify the open questions and ambiguities, and it has helped to further improve the quality of the manuscript. Only few points are left that I could be addressed in a minor revision.
- Figure 1, 4, 5: Please ensure the note is displayed at the bottom of the figure.
- Figure 5: Possibly remove lines that connect the bars. These could be misunderstood as indicating sequences, where the bars actually correspond with conditions.
- Figure 6 does not have a note. It could be mentioned that colors were all in the odd condition so readers may not wonder what has possibly caused the alternating jumps in the curve? (se point 10)
- Table 4: Possibly move JND to the title line instead of displaying it in one of the cells that should display empirical results. Bty, I did not notice SD in this table.
- Regarding point 7: Possibly add the reference for the adaptive algorithm also in the manuscript.
The language is fine.
Author Response
Dear reviewer and editor, please see the attachment.

Reviewer 2 Report (New Reviewer)
Although this is an interesting paper, it is not clear what were the examined variables. The literature review should also include the symmetry component of the stimuli (for a review on this topic see: Giannouli, V. (2013). Visual symmetry perception. Encephalos, 50, 31-42.), which is disregarded.
The reported influential studies in the introduction are not described in detail.
Is this a replication study?
The sample size is small and unjustified. Therefore, all analyses are under question about their generizability.
2.3.1.1. section is not easy to read. Please rewrite.
In the Results section the tables are chaotic.
The discussion needs to be more detailed and supported by relevant references of prior research.
Extensive language editing is needed throughout the text.
Author Response
Dear reviewer and editor, Please see the attachment.

Reviewer 3 Report (New Reviewer)
Appraisal: In the Introduction, please modify the first sentence. Indeed, it is not clear the relationship between human beings and information system. Overall, it seems to be quite misleading. According to me, the topic is interesting and it is basically related with several types of modern-day work figures.
I agree with the authors about the complexity of the topic, but I invite the authors to summarize the principal findings avoiding the detailed description of each study. Moreover, I suggest to preserve the structure of the introduction, following the previous recommendations. Furthermore, I suggest to use “Study” Instead of “Paper”. Paper is more informal than Study. The hypotheses should be written in a more explicit way. The study is complex and potentially interesting and the aims are quite reductive.
In the methods, Authors collected 32 participants. The sample size is quite small and needs to be justified. Moreover, More details are needed about the participants that not completed the task (i.e “Based on this criterion..”) and about the inclusion criteria (i.e Absence of neuropsychiatric disorders, Past history of serious illness etc.).
The Calculation of The dispersion degree of the score Ki (in the methods) uses the same terms explained in the equation (2). If it is the same, the authors should specify it. Moreover, in the Ki is not clear if 2/q-1 or all the sum is dived for q-1.
“The customized experimental program Processing was used to record data, and variance analysis was performed in SPSS. The convergence algorithm obtained different response accuracy in this experiment..” This is not clear. The term “variance analysis” is not clear. Did the authors mean “analysis of variance (ANOVA)? More, the authors need to specify the type of ANOVA that they calculated. I agree that the authors specified in a good manner the dependent variables and I suppose that all the dependent variables are continuous. Here, I advise to report if it was possible to carry out the ANOVA (factorial?, since the authors refer 2-way ANOVA). For this reason, I advise to assess homogeneity of variance and normality. Similarly, in the results, the authors reported high df values, for example F(2,2637). Please check because I think that values like these are very high. In the same way, when an effect is not significant is “not significant” and insignificant is not correct. Also, the authors can report or p=0.XXX or p<0.05 (Only an advise). Did the authors calculated post hoc comparisons?
According to me, the results should be organized in a more intelligible way.
The figure 7 should be put before in order to help the readers.
Author Response
Dear reviewer and editor, Please see the attachment.

Round 2
Reviewer 3 Report (New Reviewer)
The authors addressed all the issues that I have raised. However, the small sample size needs to be stated in the limitations. Despite the G Power calculation performed, this needs to be stated as a limitation of the study.
Author Response
Please see the attachment.

This manuscript is a resubmission of an earlier submission. The following is a list of the peer review reports and author responses from that submission.
Round 1
Reviewer 1 Report
The author presents a highly interesting study testing task difficulty and visual complexity as determinants of task performance in the Cluster Separation Task. This contribution is timely and addresses an area that has not received much attention in research. The manuscript is informative, well structured, and concisely written.
My primary concern is the way how the ANOVAs are conducted and presented. I suggest that separate tables are presented for the dependent variables, and results for factors tested in the same model should be presented in consecutive rows. Possibly it is enough also to present the joint analysis (Table 7) that tests both predictors (difficulty and complexity) as well as their interaction effect.
Further, it should be tested if the dependent variables RT and RA are negatively related. Only in this case, the computation of the composite score makes sense (i.e., speed and accuracy as indicators of performance). Conversely, if there is a speed-accuracy trade-off, it is posily better not to aggregate.
Minor issues (in chronological order)
- The Introduction is generally well structured and informative. Only some of the algorithms used to compute visual complexity are not easily comprehensible without more detailed information (cf. p. 2). However, this could stay as is, as it is not of direct relevance for this contribution.
- Figure 1: Is this figure needed? In how far does it go beyond the information already provided in the main text?
- Table 1: The header on the right side is not in boldface. By contrast, the first entry in the left column is in boldface.
- Pp. 5/6: Where there really only 8 displays in total? I assume that these are just examples of 8 different types of displays. Afterall, the experiment took 15-20 minutes to complete.
- Table 3 and 4: Please add the SD for each of the mean so the variability of the ratings can be inferred.
- P. 8, para 2: "Any point greater than two standard deviations was excluded to avoid showing dots outside the screen range and generated a new dot to re-place it": This will affect the mean of the cluster. Was the mean determined only afterwards or were points resampled until the mean (centroid) reached a certain value with given precision?
- P8, para 4: Why was an asymmetric adaptive algorithm chosen, i.e., increase by 5 pt, but decrease by 3 pt?
- P. 9, para3/4: Participants were asked "to respond as quickly as possible while making a correct judgment". This instruction will likely result in a speed-accuracy trade-off (i.e., a positive relation of AC with RT). If this was the case, the composite score CE is possibly not warranted. Personally, I would also exclude the subjective ME measure, as this captures something different from objective performance.
- ANOVA tables 5-7: I found the formatting rather unconventional: I think there shold be separate tables for the dependent variables (RA, RT, …), and the factors tested in each independently conducted ANOVA should be presented one by one in consecutive rows. What is meant by "Corrected Model", is this the intercept? I am not sure if the SS and MS columns are necessary, as they are difficult to interpret anyway.
- Personally, I would rather prefer parsimonious analyses. Hence, Tables 5 + 6 could be omitted and only Table 7 be presented instead. This is, because both effects (difficulty and complexity) are jointly tested in the last ANOVA anyway.
- Figure 7a: The regular zig-zag curve (odd encodings have larger values in all cases) is striking: What is the reason for this? Is there possibly a computational artifact that results in higher difficulty when the encoding is odd?
- P. 13, para 2: The evaluation of findings should not be in terms of statistical significance (as virtually all effects are significant anyway). Instead, the author should possibly reflect the magnitude of the effect sizes.
- P 13 / Figure 8: I found the example compelling. However, are there any other real-life situations for which the Cluster Separation Task could be predictive?
Reviewer 2 Report
This study employs a cluster separation task to investigate how visual complexity and task difficulty correlate with reaction time, reaction accuracy, mental effort and a composite measure consisting of these three response variables. I understand the potential of the study which lies in the combination of visual complexity and task difficulty, and introducing the composite measure. However, the manuscript is an extremely difficult read due to a poor structure and several language issues throughout the text, and I failed to understand what the participants were actually supposed to do in response to the displays. The participants "were required to determine which average of the two classes in the scatters was higher", but I am not sure what this means.
Furthermore, no information about basic ethical principles such as informed consent is provided, raising ethical concerns about this study.
Please see below for detailed comments.
Abstract
Already at this stage, several language issues hinder comprehension of the text.
"cognition behavior study" - the study would typically be presented as "cognitive behavioural".
"about the cluster separation task" - this is awkward and complicates the sentence.
"mental effect" - did the Authors mean "mental effort"?
Introduction
"For example, in recent years, more than 50% of civil aircraft breakups worldwide have been due to pilots' cognitive bias toward flight monitoring interface signals, leading to failed mission decisions [2]." Please rephrase, I do not understand what "cognitive bias toward flight monitoring interface signals" means.
"With the seminal work of East- man & Bertin [5], Cleveland & McGill [6], and Lewandowsky & Spence [7], recent work by Rensink & Baldridge [8], Szafir [9], and Shen et al. [10]." - something is wrong with this sentence.
"In between, visual task..." - I do not understand the "In between" part.
"Therefore, to help organize the discussion at this interface, In the present study" - please revisit this sentence.
Methods
"Observers" - these are typically called "Participants". I do not understand why the Authors have not used a well-established term.
"Apparatus" - how was the procedure delivered, i.e., with what software?
General discussion
Figure 8 should be introduced in the Introduction and then discussed here in relation to the results.